# Exploring the Effects of Cancer as a Traumatic Event on Italian Adolescents and Young Adults: Investigating Psychological Well-Being, Identity Construction and Coping Strategies

Chiara Ionio [1,2,*], Francesca Bigoni [3,4], Maddalena Sacchi [1], Marco Zecca [4], Elena Bergami [4], Marta Landoni [1], Giulia Ciuffo [1,2], Anna Rovati [3,4] and Damiano Rizzi [3,4]

1 CRIdee, Unità di Ricerca sul Trauma, Dipartimento di Psicologia, Università Cattolica, 20123 Milano, Italy
2 Società Italiana di Psicologia Pediatrica (S.I.P.Ped), 90144 Palermo, Italy
3 Fondazione Soleterre, 20125 Milano, Italy
4 Fondazione Policlinico San Matteo IRCCS, 27100 Pavia, Italy
* Correspondence: chiara.ionio@unicatt.it; Tel.: +39-027-234-3642 or +39-338-442-5218; Fax: +39-027-234-2280

**Abstract:** Cancer in adolescence is considered a family disease that can have numerous negative psychological consequences for adolescents and the entire household. The aim of this study was to investigate the impact of oncological disease in adolescence, with particular reference to the psychological and post-traumatic consequences for the adolescents themselves and the family system. An explorative case–control study was conducted with 31 adolescents (mean age 18.03 ± 2.799) hospitalised for cancer at IRCCS San Matteo Hospital in Pavia and 47 healthy adolescents (mean age 16.17 ± 2.099). The two samples completed a survey that included sociodemographic information and questionnaires assessing psychological well-being, traumatic effects of the disease, and adequacy of the relationship with parents. 56.7% of oncology adolescents scored below average in psychological well-being, and a small proportion of them fell within the range of clinical concern for anger (9.7%), PTS (12.9%), and dissociation (12.9%). Compared with peers, there were no significant differences. However, in contrast to peers, oncology adolescents showed a strong influence of the traumatic event on the construction of their identity and life perspectives. A significantly positive correlation also emerged between adolescents' psychological well-being and the relationship with their parents (mothers: r = 0.796; p < 0.01; fathers: r = 0.692; p < 0.01). Our findings highlight how cancer in adolescence could represent a central traumatic event that can shape the identity and life of teenagers who are in an intrinsically delicate and vulnerable stage of life.

**Keywords:** adolescence; cancer; trauma; post-traumatic stress; psychological well-being; parent-child relationship





## 1. Introduction

In 2020, approximately 10 million people worldwide will have died from cancer, making it one of the leading causes of death [1]. A total 400,000 children and adolescents aged 0 to 19 years are diagnosed with cancer each year, and the incidence of the disease tends to increase with age (Lam et al., 2019; World Health Organization, 2021).

The third volume of the International Childhood Cancer Incidence (IICC-3) published data showing worldwide differences in overall cancer incidence by geographic region, with southern Europe having the highest rates in the 15- to 19-year-old age group [2,3]. With a standardized incidence rate of 275.4 per million person-years for the 15- to 19-year-old age group, data collected by the Italian Network of Cancer Registries (AIRTUM) from 1992 to 2013 from the 26 local Italian cancer registries confirmed that Italy is one of the European countries with the highest incidence rates [2,3]. Recently, the Italian Association of Cancer Registries (AIRTUM) estimated that 4000 cases will be diagnosed in adolescents

(15–19 years) in Italy between 2016 and 2020. The estimated annual average is 1400 cases in the 0–14 years age group and 900 cases in the 15–19 years age group [4].

However, cancer diagnosis and treatment can cause significant psychological and emotional distress (Akimana et al., 2019). Compared to the adult population, adolescent patients present with different cancer patterns with unique biological and psychological characteristics. In fact, adolescence is already characterized by physical, intellectual, affective, and social changes that are of great importance to individuals [5].

Since puberty itself is a very complex and delicate developmental period, an illness occurring during this time can be a traumatic experience for the child and the entire family, with many short- and long-term psychological consequences [6]. Post-traumatic stress disorder (PTSD) is described as "a psychological disorder that can occur in individuals who have experienced or witnessed a horrific event" (American Psychiatric Association, 2020). When considering the perception of cancer as trauma in adolescence, it is also important to consider the risk factors that may contribute to additional stress and perception of trauma, such as intense and painful treatments, hospitalization, and daily separation from family and friends [7]. In addition, the neoplasm represents another challenge that is added to normal developmental tasks [8]. For example, during the disease, young people experience the side effects of treatment: hair and eyelash loss, surgical scars, weight changes, and eventually persistent fatigue. These threaten the young person's self-image and self-esteem [9].

Recently, the pandemic COVID-19 has raised additional difficulties. The COVID-19 pandemic has placed a heavy burden on health care systems and has led to widespread disruptions in cancer care. This has led to delays in the diagnosis and treatment of cancer patients [10–12]. In addition, cancer patients have a higher risk of experiencing complications due to their weakened immune system and pre-existing conditions related to their cancer and its treatment COVID-19 [13]. The impact of changing dynamics in cancer care is felt by adolescents and young adults (AYAs) aged 15 to 39 years; however, their unique developmental, educational, social, and emotional needs may make them more vulnerable to the negative effects of this pandemic [14].

However, research has shown that there are both negative and positive effects of cancer. Many people who have undergone cancer treatment have reportedly expressed positive feelings about their cancer diagnosis [5]. Post-traumatic growth (PTG) can be defined as the cognitive process undertaken to make sense of a traumatic event by reinterpreting the traumatic event in a positive way [5,7]. The role of coping strategies is fundamental in this process.

According to a recent review, knowledge of how children and adolescents cope with chronic illnesses such as cancer has generally improved [15]. Three coping strategies can be distinguished: primary control coping, secondary control coping, and disengagement coping. These strategies are based on the model of perceived control in children and adolescents developed by Weisz and colleagues [16–20]. Problem solving or changing one's emotional responses to the stressor are examples of primary control coping techniques (e.g., emotional expression and emotional modulation). Attempts to manage stress are referred to as secondary control (e.g., cognitive reappraisal, positive thinking, acceptance). In addition, last but not least, disengagement coping involves attempts to divert attention from the cause of the stress or one's reactions to it (e.g., avoidance, denial, wishful thinking).

Of all the coping strategies, the most important one highlighted in the literature is social support from parents, friends, and health care providers [21], depending on the developmental stage. An important aspect of an adolescent's life is belonging to a peer group. Close relationships with peers are also an important source of support for chronically ill adolescents at a time when they are coping with both developmental tasks and illness-related challenges. While parents continue to play the role of primary caregivers, friends and peers provide emotional support by accepting their ill friend with his or her physical limitation.

## 2. Materials and Methods

*2.1. Data Collection*

This study was conducted by the Soleterre Onlus Foundation, the Trauma Psychology Research Unit of the Catholic University of Milan, and the IRCCS San Matteo Hospital in Pavia to investigate risk and protective factors for quality of life and psychological and relational adjustment in children and adolescents with a cancer diagnosis.

The study was approved by the Ethics Committee of IRCCS San Matteo Hospital in Pavia, Italy (prot. no. 20200063196). Minor participants were required to provide informed consent signed by both parents. Adult participants were asked to sign the same informed consent. The survey took place from October 2019 to October 2021, with an interruption during the health emergency due to the pandemic COVID-19.

The survey was administered to adolescents with oncological disease by Soleterre staff at the Paediatric Oncohematology Unit of the IRCCS Policlinico San Matteo Foundation in Pavia.

The survey of participants in the control sample, on the other hand, began in November 2021 and ended in March 2022. Healthy adolescents were recruited in schools and answered the questionnaire online through the Qualtrics platform.

All data collected were entered into a dedicated database, with participants identified only by a unique ID number. The database was stored on a secure server, and access to the information was limited to members of the research team.

*2.2. Participants*

Our clinical sample consisted of 31 adolescents and young adults aged 13 to 24 years (18.03 ± 2.799) who had been diagnosed with cancer and were undergoing treatment at the Paediatric Oncohematology Unit of the IRCCS Policlinico San Matteo Foundation in Pavia.

The group consisted of 54.8% males and 45.2% females. Most of them (73.3%) were attending secondary school. Of the group, 45.2% reported living with their parents and siblings, 16.1% lived with their parents, and the remaining 38.7% reported living with only one parent (mother or father). The most common diseases represented in the sample were Hodgkin's lymphoma (14.3%) and acute lymphoblastic leukemia (LLA) (14.3%). Inclusion criteria were both sexes, with an oncological disease, with a good knowledge of Italian, and recruited from the Department and Day Hospital of Oncohematology.

Our control group consisted of 47 adolescents aged 11 to 18 years (16.17 ± 2099). It consisted of 36.2% males and 63.8% females. Most of them (74.5%) were attending secondary school. Regarding the clinical sample, most (68.1%) reported living with their parents and siblings, 25.5% lived with their parents, and the remaining 6.4% reported living with only one parent (mother or father). Inclusion criteria were both sexes, without oncological disease, with a good knowledge of the Italian language.

*2.3. Ethics*

All study procedures were reviewed and approved by the Ethics Committee of the IRCCS Policlinico San Matteo Foundation in Pavia (protocols no. 20200063196). Informed consent was obtained from all subjects involved in the study. Data were collected in accordance with the principles of the Declaration of Helsinki and in compliance with the IRB ethical guidelines.

*2.4. Measures*

The adolescents and young adults were asked to complete the following questionnaires:

1. TRI. Test of Interpersonal Relations (Bracken, 1993; Italian validation by Inaes, 1996): It was designed to assess the adequacy of children's interpersonal relationships in the social domain, i.e., in relation to peers, in school, in relation to teachers, and in the family in relation to the relationship with parents. It consists of 35 items, each rated on a 5-point Likert scale from "strongly agree" to "strongly disagree". In this work, we specifically used scales related to adolescents' perceptions of the quality of

their relationship with their mother and father. The questionnaire had good internal consistency (Cronbach's alpha ranged from 0.93 to 0.96).

2. KIDSCREEN-27 (Italian validation by The KIDSCREEN GROUP, 2004): Allows the assessment of well-being and health related to quality of life. It consists of 27 items and measures five Rasch-scaled dimensions: (1) Physical well-being, (2) Psychological well-being, (3) Autonomy and relationship with parents, (4) Peers and social support, (5) School environment. Each item is scored on a 5-point Likert scale ranging from 1 for "not at all" to 5 for "very much". Higher scores indicate better quality of life and social support. Construct validity was assessed by calculating Cohen's effect size (ES = 0.54). The questionnaire had good internal consistency (Cronbach's alpha > 0.70).

3. Centrality of Events Scale (CES; Berntsen and Rubin, 2006; Italian validation by Ionio, Mascheroni, and Di Blasio, 2018): a self-report measure designed to assess the extent to which the memory of a stressful and traumatic event was central to one's (a) life history, (b) personal identity, and (c) attribution of meaning to other personal life events. These three factors are assessed using 20 items on a 5-point Likert scale ranging from 1 for "strongly disagree" to 5 for "strongly agree". The questionnaire has good internal consistency (Cronbach's alpha = 0.94).

4. Trauma Symptom Checklist for Children (TSCC-A; Briere, 2011; Italian validation by Di Blasio, Piccolo, Traficante, 2011): used to assess post-traumatic stress and related psychological symptoms. This instrument is particularly suitable for assessing children and adolescents aged 11 to 16 years who have experienced traumatic events. Each item is rated on a 4-point Likert scale ranging from 0 for "never" to 3 for "almost always". We used the 44-item version, which does not include references to sexual issues. The questionnaire consists of the following five clinical scales: (1) Anxiety, which captures general fear, overexcitement, worry, specific fears (e.g., of men, women, or both, of the dark, of being killed), episodes of free-floating fear, and a sense of impending danger. (2) Depression: feelings of sadness, unhappiness, and loneliness, episodes of weepiness, depressive cognitions such as guilt and self-denial, and self-harm and suicidality. (3) Anger, which deals with angry thoughts, feelings, and behaviors, such as feeling angry, being mean and hating others, having difficulty de-escalating anger, wanting to yell at or hurt people, arguing, or fighting. (4) Post-traumatic stress, which captures post-traumatic symptoms such as intrusive thoughts, sensations, and memories of painful past events, nightmares, anxiety, and cognitive avoidance of painful feelings; and (5) Dissociation, which examines dissociative symptoms such as derealization, thought emptiness, emotional numbing, pretending to be another person or place, daydreaming, memory problems, and dissociative avoidance. We chose to use the TSCC-A because our clinical sample was not in treatment, and this may have influenced the results of the Sexual Concerns scale. The questionnaire showed good internal consistency (Cronbach's alpha = 0.83).

*2.5. Analysis*

SPSS Statistics version 27.0 was used to analyze the data. Descriptive analyses were performed first to examine the sociodemographic characteristics of the sample. The t-test for independent samples was used to compare the two groups in terms of psychological well-being and possible traumatic effects of the disease. Pearson correlations were used to examine the association between an appropriate relationship with parents and adolescents' psychological well-being.

**3. Results**

*3.1. Psychological Well-Being and Effects of the Traumatic Event on Adolescents*

To examine the psychological well-being of the adolescents in our sample, we converted the raw scores of the KIDSCREEN-27 into T-scores, with a mean of 50 and a standard deviation of 10, based on the parameters of the Rasch person. The results of the descriptive analysis performed show that the mean score of our subjects in relation to oncological

adolescents was 48.64 (SD 10.30). In fact, 56.7% of our subjects scored below average for psychological well-being. In the control sample, the average score of our subjects was 50.91 (SD 9.80). Only 44.5% of our sample had below average scores for psychological well-being. We then compared the scores of our two samples using independent samples t-tests and found no significant differences in psychological well-being (t = −0.954; *p* = 0.344; gl = 60.111). Using the Trauma Symptom Checklist for Children, we then specifically examined anxiety, depression, anger, post-traumatic stress, and dissociation. Most subjects in the clinical sample were within the normal range on all five clinical scales (anxiety 87.1%; depression 96.8%; anger 90.3%; PTS 83.9%; dissociation 83.9%). However, a small portion of the sample fell within the range of clinical concern (Anger 9.7%; PTS 12.9%; Dissociation 12.9%). The results of the control sample show that most of them fell within a normal range (anxiety 89.4%; depression 85.1%; anger 80.9%; PTS 87.2%; dissociation 80.9%). In contrast, a small proportion of the sample had clinically worrisome scores (depression 8.5%; anger 12.8%; PTS 10.6%; dissociation 10.6%). We then compared the results of our two samples of adolescents using independent samples t-tests and found a significant difference on the clinical depression subscale of the TSCC (t= −2.092; *p* = 0.040; gl = 69.752).

To examine the centrality of the traumatic event (oncologic disease) in the adolescents' lives, we converted the raw scores of the CES into T-scores, with a mean of 50 and a standard deviation of 10, based on the Rasch person parameter. For oncology adolescents, the mean score of the first subscale (Factor I) was 53.36 (SD 8.98). Most of the sample (75.8%) scored above average on the impact of traumatic memories on daily life (Factor I). The average score of these adolescents in the second subscale (Factor II) was 55.86 (SD 8.11). 72.2% of the oncology sample scored above average on the impact of traumatic memories on personal identity (Factor II). In addition, their mean score on the third subscale (Factor III) was 54.39 (SD 9.88). 68.8% of the clinical sample scored above average on the impact of traumatic memories on life perspective. The average total score (impact of traumatic memories) of our adolescents was 55.24 (SD 8.42), and 79.6% of the sample scored above average. In contrast, the average score of our subjects in the control group in the first subscale (Factor I) was 47.88 (SD 10.11).

Most of the sample (52%) scored below average on the impact of traumatic memories on daily life (Factor I). The average score of these adolescents on the second subscale (Factor II) was 46.30 (SD 9.34). 69.7% of the adolescents scored below average on the impact of traumatic memories on personal identity (Factor II). (Factor II). In addition, their mean score on the third subscale (Factor III) was 47.23 (SD 9.13). 60.8% of our adolescents scored below average on the impact of traumatic memories on life perspective. The average total score (impact of traumatic memories) of our adolescents was 46.69 (SD 9.55) and 69.5% of them scored below average. We then compared the results of the two samples using independent samples t-tests. As shown in Table 1, the results showed significant differences between the two groups for all subscales and the total scale.

**Table 1.** CES t-test for independent samples.

| CES | Sample | Mean | Standard Deviation | T | Sign. | gl |
|---|---|---|---|---|---|---|
| Factor I | Clinical group | 53.36 | 8.98 | 2.450 | 0.017 | 64.873 |
| | Control group | 47.88 | 10.11 | | | |
| Factor II | Clinical group | 55.86 | 8.11 | 4.686 | 0.000014 | 65.742 |
| | Control group | 46.30 | 9.34 | | | |
| Factor III | Clinical group | 54.39 | 9.88 | 3.146 | 0.003 | 56.123 |
| | Control group | 47.23 | 9.13 | | | |
| Total | Clinical group | 55.24 | 8.42 | 4.061 | 0.000134 | 65.161 |
| | Control group | 46.69 | 9.55 | | | |

### 3.2. Effects of the Traumatic Event on the Family System

To examine the adequacy of the relationship between adolescents with cancer and those without cancer and their parents, we used the Interpersonal Relationship Test (TRI). In the oncology sample, most adolescents had an average relationship with both their mother and father (41.9% and 51.6%, respectively). Of the sample, 38.7% had a positive relationship with their mother, while only 16.1% had a positive relationship with their father.

In the control group, most adolescents had an average relationship with their mother and father (36.2% and 44.7%, respectively). Of the control group, 27.7% of adolescents had a positive relationship with their mother and 29.8% with their father, while 19.1% of adolescents had a negative relationship with both. The results of descriptive analysis of TRI are shown in Figure 1.

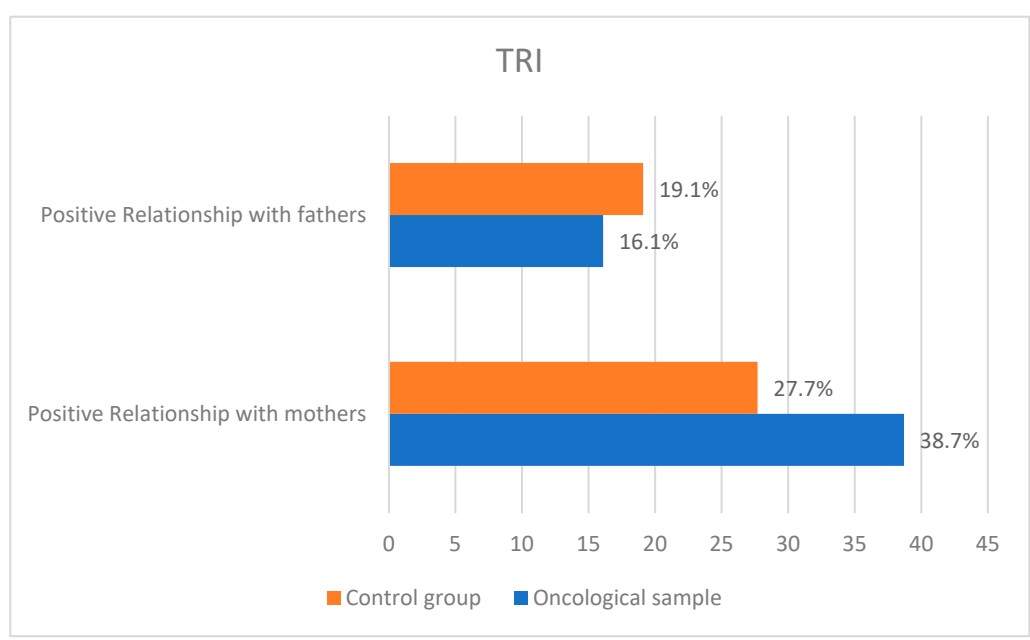

**Figure 1.** Results of TRI's descriptive analysis for the two samples.

In addition, the results of the correlation analyses performed show that there is a significant positive relationship between the adolescents' psychological well-being (Kidscreen-27) and their relationship with their parents (TRI) (mothers: r = 0.796; *p* < 0.01; fathers: r = 0.692; *p* < 0.01).

## 4. Discussion

The purpose of this study was to examine psychological well-being and parent-child relationships in adolescents and young adults with cancer compared with healthy peers, focusing on how cancer is perceived as a traumatic event.

### 4.1. Psychological Well-Being and Effects of the Traumatic Event on Adolescents

While research on psychological well-being and quality of life during cancer treatment has focused primarily on childhood cancer survivors [22], studies examining adolescents' and young adults' psychological adjustment during treatment have yielded inconsistent results. The results of analyses conducted on our sample of adolescents and young adults partially confirm the findings of the recent literature. A significant proportion of our clinical sample exhibited below-average psychological well-being, indicating a lack of positive emotions and life satisfaction, as well as a low experience of positive emotions such as happiness, joy, and cheerfulness. In addition, a small but still consistent proportion of adolescents fell into the range of clinical concerns for anger, PTS, and dissociation. These findings appear to be consistent with the work of Briere [23], suggesting that there are many

different maladaptive outcomes that can occur as a result of cancer. However, contrary to expectations, we found no significant differences between the psychological well-being of adolescents with cancer and that of their peers, with the exception of depression. Some studies suggest that the majority of these patients do not have significantly higher levels of anxiety or depression compared with their peers [24]. However, a minority of them (17–30%) were found to have symptoms of depression and anxiety [25]). There are several possible explanations for the lack of significant differences in psychological well-being between our samples of adolescents and AYA. First, the samples are unfortunately not perfectly balanced in terms of age and size. In addition, previous research has found that youth resilience increases as a result of their trauma exposure, leading to better adjustment during the process [26]. Indeed, cancer may have led to post-traumatic growth in these adolescents, attenuating the differences between the two samples. In addition, we need to consider that the COVID-19 pandemic also has implications for the mental health of adolescents and young adults whom we consider "healthy". Healthy adolescents and young adults may have reported more symptoms during this time because COVID-19 affected their lives more than patients who are already accustomed to being isolated because of their illness.

Finally, our findings underscored the central role of cancer as a traumatic event in the personal identity and life history of adolescents compared with peers, suggesting that this event may have maladaptive psychological consequences that are also felt in the medium and long term. These findings are consistent with previous research [27,28] reporting how many adolescents and AYA described cancer or cancer-related events as the most stressful and traumatic experiences of their lives. These data highlight the importance of mental health professionals paying attention not only to the physical health but also to the mental health of these young patients who are facing the disease at such a sensitive time in their development.

### 4.2. Effects of the Traumatic Event on the Family System

The literature suggests that adolescent illness can lead to withdrawal and ambivalence toward parents, while parents remain an important protective factor during this sensitive life event [7]. The results of our clinical sample seem to confirm these findings, as most adolescents and young adults have an adequate relationship with their parents. The fact that the relationship with the mother is more positive could be due to her role as the main caregiver. Indeed, the mother often takes time off from work or chooses a part-time solution so that she can devote all her time to her child [29]. As expected, most healthy adolescents and AYA had an average relationship with both parents. The latter could be explained by the ambivalence typical of this stage of life, as adolescents begin a process of emancipation from the family to gain autonomy and independence [5]. Moreover, our correlation analysis revealed a positive association between adolescents' psychological well-being and their relationship with their parents, suggesting that parental support may help mitigate the negative effects of cancer and positively influence adolescents' well-being [30,31]. In addition, separating these young patients from their families due to hospitalization at a time when they are very vulnerable may contribute to increasing their dependence and desire for closeness with parental caregivers, which is not usually the case at this developmental stage [8]. These findings shed light on the protective role of parental involvement and support for adolescents and young adults diagnosed with cancer and underscore the importance of parental involvement in treatment by health care providers.

### 5. Limitations

Despite the significant results, we cannot overlook some limitations of the present study. First, our samples were quite small, and not perfectly balanced in terms of age or size. Further studies are needed to examine differences and potential risk factors related to psychological well-being in adolescents and young adults with cancer compared with their peers. In addition, it is imperative to keep in mind that these data were collected during

the COVID-19 pandemic. Therefore, as mentioned earlier, it is important to recognize that healthy adolescents and young adults were also psychologically affected by the pandemic. They may even have experienced more symptoms than those who were already accustomed to disease-induced isolation, which may have partially influenced our results.

## 6. Conclusions

The research hypotheses of the present study were well supported by the analysis performed and, despite the current limited sample size, provide important support for what is already known on this topic. Our findings highlight that cancer in adolescence may represent a key traumatic event that can shape the identities and lives of teens who are at an inherently delicate and vulnerable stage of life. These findings underscore the importance of timely and targeted intervention by clinicians and therapists. As we mentioned earlier, hospitalization often causes these teens to feel isolated from friendships and the peer group at a developmental stage when peers play a critical role. For this reason, it can be beneficial for teens and young adults to be part of a support group where they can meet others who are going through similar experiences, share their problems, and find comfort in knowing that they are not alone. In addition, the hospital school could help maintain that sense of normalcy during a difficult time in their lives. The opportunity to attend school and interact with peers may help adolescents maintain social skills and avoid feelings of isolation. In addition, our findings suggest the protective role of parents in mitigating the psychological consequences of the disease. These findings provide useful guidance for designing interventions that involve parents and target the well-being of the entire family system, not just the individual adolescent. Indeed, it is critical to involve parents in the treatment planning process to ensure that treatment is tailored to the specific needs of the family, which could also help the family feel more involved and engaged in the process. In addition, therapists should provide education and support to family members to help them better understand what the adolescent is going through and provide them with strategies for honest and effective communication with the adolescent.

**Author Contributions:** Conceptualization and investigation D.R., M.Z., F.B., M.S., E.B., A.R. and C.I.; methodology C.I., G.C. and M.L.; validation, formal analysis and Writing—original draft preparation, C.I., G.C. and M.L.; writing—review and editing, D.R., C.I., G.C. and M.L. All authors have read and agreed to the published version of the manuscript.

**Funding:** This research received no external funding.

**Institutional Review Board Statement:** The study was approved by the Ethics Committee of IRCCS San Matteo Hospital in Pavia, Italy (prot. no. 20200063196).

**Informed Consent Statement:** Informed consent was obtained from all subjects involved in the study.

**Data Availability Statement:** The dataset in not publicly available due to the hospital privacy policy.

**Acknowledgments:** Fondazione Soleterre, IRCCS San Matteo Hospital.

**Conflicts of Interest:** The authors declare no conflict of interest.

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
