# Peer review of "Exploring the Effects of Cancer as a Traumatic Event on Italian Adolescents and Young Adults: Investigating Psychological Well-Being, Identity Construction and Coping Strategies"

_pediatrrep, doi:10.3390/pediatric15010021_

Round 1

Reviewer 1 Report

This study focused on psychological wellbeing, interpersonal relationship with parents and trauma in AYA with cancer compared with healthy peers. The topic is quite new and could be valuable to have more information on this area.

However, I think to have some suggestions that could be make the paper more interesting and scientific.

Title: It doesn’t respect all the issues examined in the study. Try to reformulate it.

Introduction

It should be more extensive. Add for example the results of other articles such as: Tremolada M., Taverna L. Chiavetta I.T., Bonichini S., Putti M.C., Biffi A. (2020). Psychological wellbeing in adolescents with leukaemia: A comparative study with typical development peers, Int. J. Environ. Res. Public Health, 17, 567 doi:10.3390/ijerph17020567.

Methods

Participants should be defined as adolescents and young adults basing on their age range: 13-24

Correct line 111.

The number and age of the control group is relatively different form the clinic group. This is to be considered in the interpretation of your analyses.

At what time point of treatment were the data collected? It is important to now it to understand also the results of comparison with the control group.

It should be motivated why you decided to not include the sexual themes of the TSCC. The different scales of the questionnaire should be showed.

How the covid pandemic could influence your data? Discuss it in the limits section.

Have you run analyses on the possible differences between the two groups on TRI? I think that the positive or negative percentages on relationship with adolescents’ parents could be significantly different.

Discussion

A result showed no significant differences between the psychological well-being of adolescents with cancer and their healthy peers and some explanations were given. Could be also the particular period of Covid pandemic to alter this result? Healthy AYA could report more symptomatology in this period with covid19 impacting more on their lives than patients that are already get used to be isolated for their illness.

Author Response

Dear reviewer,

We revised the manuscript entitled “Cancer and adolescence: An exploratory study on the traumatic impact of an oncological disease on Italian adolescents”, submitted to Pediatric Reports.

We thank you for taking the time for reviewing our article. 

We are grateful for the meaningful suggestion you gave us.

We have followed your suggestions, trying to answer all comments.

You can find our response for each point highlight below.

Comment #1: This study focused on psychological wellbeing, interpersonal relationship with parents and trauma in AYA with cancer compared with healthy peers. The topic is quite new and could be valuable to have more information on this area. However, I think to have some suggestions that could be make the paper more interesting and scientific.

Reply#1: We thank the reviewer for the positive comment.

Comment #2: Title: It doesn’t respect all the issues examined in the study. Try to reformulate it.

Reply#2: we reformulated it.

Comment #3: Introduction: It should be more extensive. Add for example the results of other articles such as: Tremolada M., Taverna L. Chiavetta I.T., Bonichini S., Putti M.C., Biffi A. (2020). Psychological wellbeing in adolescents with leukaemia: A comparative study with typical development peers, Int. J. Environ. Res. Public Health, 17, 567 doi:10.3390/ijerph17020567.

Reply#3: We followed the advice and added in the introduction.

Comment#4: Participants should be defined as adolescents and young adults basing on their age range: 13-24. Correct line 111.

Reply#4: we followed your advice and corrected the wording.

Comment#5: The number and age of the control group is relatively different form the clinic group. This is to be considered in the interpretation of your analyses.

Reply#5: we followed the advice and added that information in discussion and limitations section.

Comment#6: At what time point of treatment were the data collected? It is important to now it to understand also the results of comparison with the control group.

Reply#6: we followed your advice and provided that information in participants’ section.

Comment#7: It should be motivated why you decided to not include the sexual themes of the TSCC. The different scales of the questionnaire should be showed.

Reply#7: we explained it in measures section and showed the different scales as suggested.

Comment#8: How the covid pandemic could influence your data? Discuss it in the limitations section.

Reply#8: we followed your advice.

Comment#9: Have you run analyses on the possible differences between the two groups on TRI? I think that the positive or negative percentages on relationship with adolescents’ parents could be significantly different.

Reply#9: we’ve run a t-test for independent samples, but no significant differences emerged.

Comment#10: A result showed no significant differences between the psychological well-being of adolescents with cancer and their healthy peers, and some explanations were given. Could be also the period of Covid pandemic to alter this result? Healthy AYA could report more symptomatology in this period with covid19 impacting more on their lives than patients that are already get used to be isolated for their illness.

Reply#10: we added this information in discussion and limitations section.

Reviewer 2 Report

  1-The Introduction contains a lot of details, making it too lengthy. Some information in the Information is also redundant to a number of statements in the Results and Discussion sections. Please make the Introduction shorter

2-Some explanations on the part of the measure will be appreciated.

Author Response

Dear reviewer,

We revised the manuscript entitled “Cancer and adolescence: An exploratory study on the traumatic impact of an oncological disease on Italian adolescents”, submitted to Pediatric Reports.

We thank you for taking the time for reviewing our article. 

We are grateful for the meaningful suggestion you gave us.

We have followed your suggestions, trying to answer all comments.

You can find our response for each point highlight below.

Comment #1: The Introduction contains a lot of details, making it too lengthy. Some information in the Information is also redundant to a number of statements in the Results and Discussion sections. Please make the Introduction shorter.

Reply#1: We followed the advice.

Comment #2: Some explanations on the part of the measure will be appreciated.

Reply#2: we followed your advice and gave more detail about measures.

Reviewer 3 Report

This study is important as it looks at the negative psychological consequences for adolescents and the closer social network. 

Intro:

- Theoretical Backdrop is missing and needs to be added, e.g., relating to self-image and self-esteem, trauma and coping.

- references for lines 73ff need to be added. 

-Add more recent references from 2021 and 2022 (and maybe 2023!)

Method:

It is great that you collected a clinical sample and a control group. But the latter one is not well matched. You need to collect individuals of the same age, gender etc., or collect a larger sample and then do propensity score matching!

if you have a hypothesis with "affected" and "impact" you need longitudinal data or even better experimental data! Eigther you need to add that or you need to revise your hypothesis and the wording of your whole article!

Explain how exactly you want to test your hypothesis, the current description is vague and how you intent to treat missings remains open.

Results: 

Run a MANOVA, controlling for confounding variables (in addition/instead of the t-tests), and a regression analysis in addition/instead of the binary correlation analyses.

You refer to "figure 2" but I do not see it in the manuscript. Also where is figure 1?

Please add at least 1 figure to communicate your results.

Discussion

needs extensive revision

-Add more recent references from 2021 and 2022 (and maybe 2023!)

Author Response

Dear reviewer,

We revised the manuscript entitled “Cancer and adolescence: An exploratory study on the traumatic impact of an oncological disease on Italian adolescents”, submitted to Pediatric Reports.

We thank you for taking the time for reviewing our article. 

We are grateful for the meaningful suggestion you gave us.

We have followed your suggestions, trying to answer all comments.

You can find our response for each point highlight below.

Comment #1: This study is important as it looks at the negative psychological consequences for adolescents and the closer social network. 

Reply#1: We thank the reviewer for the positive comment.

Comment #2: Theoretical Backdrop is missing and needs to be added, e.g., relating to self-image and self-esteem, trauma and coping.

Reply#2: We have added literature about this.

Comment #3: references for lines 73ff need to be added. 

Reply#3: We have added literature about this.

Comment #4: Add more recent references from 2021 and 2022 (and maybe 2023!)

Reply#4: We have added literature about this.

Comment #5: It is great that you collected a clinical sample and a control group. But the latter one is not well matched. You need to collect individuals of the same age, gender etc., or collect a larger sample and then do propensity score matching! if you have a hypothesis with "affected" and "impact" you need longitudinal data or even better experimental data! Eigther you need to add that or you need to revise your hypothesis and the wording of your whole article! Explain how exactly you want to test your hypothesis, the current description is vague and how you intent to treat missings remains open.

Reply#5: we followed the advice and reworded the paper.

Comment #6: Run a MANOVA, controlling for confounding variables (in addition/instead of the t-tests), and a regression analysis in addition/instead of the binary correlation analyses.

Reply#6: We thank rewiever for the comment, but both for MANOVA and regression, it is not recommended to use small groups.

Comment #7: You refer to "figure 2" but I do not see it in the manuscript. Also where is figure 1?

Reply#7: we have corrected the mistake.

Comment #8: Please add at least 1 figure to communicate your results.

Reply#8: we followed your advice.

Comment #9: discussion needs extensive revision

Reply#9: we followed the advice and revised that section.

Round 2

Reviewer 3 Report

The paper still requires improvements, e.g. 

- if you intent to investigate "impact" you need a longitudinal or experimental study

-mothers: r=.996 indicates that there is a too high overlap. Please recalculate

-Fig 1 needs the indication of standard deviations, too 

Author Response

Dear Reviewer,

We revised the manuscript entitled “Cancer and adolescence: An exploratory study on the traumatic impact of an oncological disease on Italian adolescents”, submitted to Pediatric Reports. We thank tyou for taking the time for reviewing our article further. We are grateful for the meaningful suggestion you gave us and we have followed the editor’s and reviewers’ suggestions, trying to answer all comments.

You can find our response for each point highlight below.

Comment #1: if you intent to investigate "impact" you need a longitudinal or experimental study

Reply#1: we followed your advice and reworded the paper.

Comment #2: mothers: r=.996 indicates that there is a too high overlap. Please recalculate

Reply#2: we fixed it.

Comment #3: Fig 1 needs the indication of standard deviations, too 

Reply#3: the values in the figure express percentage frequencies. We have made this clearer.

Comment #4: English language and style are fine/minor spell check required

Reply#4: We had proofread the manuscript.